# Recent Advances in Biologically Active Coumarins from Marine Sources: Synthesis and Evaluation

**DOI:** 10.3390/md21010037

**Published:** 2022-12-31

**Authors:** Laura Fernández-Peña, Maria João Matos, Enol López

**Affiliations:** 1Department of Organic Chemistry, University of Valladolid, Campus Miguel Delibes, 47011 Valladolid, Spain; 2Departamento de Química Orgánica, Facultade de Farmacia, Universidade Santiago de Compostela, 15782 Santiago de Compostela, Spain

**Keywords:** coumarin, biological activity, marine natural products, heterocycles

## Abstract

Coumarin and its derivatives have significantly attracted the attention of medicinal chemists and chemical biologists due to their huge range of biological, and in particular, pharmacological properties. Interesting families of coumarins have been found from marine sources, which has accelerated the drug discovery process by inspiring innovation or even by the identification of analogues with remarkable biological properties. The purpose of this review is to showcase the most interesting marine-derived coumarins from a medicinal chemistry point of view, as well as the novel and useful synthetic routes described to date to achieve these chemical structures. The references that compose this overview were collected from PubMed, Mendeley and SciFinder.

## 1. Introduction

The ocean can be considered the cradle of living organisms and the first and biggest ecosystem of our planet. The diversified environments found in different oceanic zones have been a matter of study since ancient times, as marine natural products have been used in functional foods, dietary supplements and medicine. The interest in marine pharmacology to discover organic molecules with active pharmaceutical properties clearly supposes a renaissance in drug development based on natural products due to the chances to increase the chemical space from this underexplored ecosystem. Compared with terrestrial substances, marine substances have different chemical features with sometimes better mechanisms of defense. It is very likely that, under the extreme conditions (e.g., low levels of oxygen, high pressures, absence of light) in which deep sea organisms live, they have adapted to this environment undergoing biochemical or physiological transformations to survive, providing sophisticated organic skeletons with interesting drug-like properties [1,2].

On the other hand, the design and synthesis of molecules in the drug discovery process is a real challenge for a medicinal chemist, as a vast number of new chemical entities have to be prepared from hit identification (Hit Id.) to a late lead optimization (LLO) process. It is estimated that approximately more than 12 years with a cost of more than one billion EUR are needed to bring a new drug to the market [3]. An alternative pathway in drug discovery programs has been the identification of “advanced chemical entities” which can provide valuable information against a molecular target. In this regard, the development of drugs based on natural product structures has been investigated for decades, providing a clear way to reduce both time and cost. Considering marine sources, around 3,000 entities with biological properties have been identified [4], which would clearly facilitate drug discovery programs in the search for new therapies [5,6,7].

Coumarin is considered a privileged scaffold that has extensively attracted the attention of the scientific community in a wide range of applications [8,9,10,11,12]. Chemically, coumarins belong to the family of lactones (2*H*-1-benzopyran-2-one, 1,2-benzopyrone or benzo-α-pyrone) and consist of fused benzene and α-pyrone rings. A general classification considering its structure would be simple coumarins, furanocoumarins, pyranocoumarins, dihydrofurano coumarins, phenylcoumarins and bicoumarins. 

They have been used in many research areas, such as cosmetics, food additives, fluorescent probes and laser dyes, among others [13]. However, coumarin derivatives have created a major impact in medicinal chemistry, where most of these derivatives have shown interesting pharmacological properties including anticoagulant, anti-inflammatory, antiviral, antioxidant, anticancer and inhibitory of enzymes [14,15,16,17,18,19]. Thus, they have been used in a variety of biological targets. For instance, acenocoumarol (**1**) and warfarin (**3**) are representative approved drugs with anticoagulatory activity, whose clinical results were studied on 2111 MPHV patients [20]. Hydroxycoumarins show promising results in the treatment of inflammation [21,22], and coumarin–chalcone and coumarin–resveratrol hybrids were studied in the treatment of neurodegenerative diseases [9,23,24]. In addition, some anticancer activities against HepG2 and HeLa cancer cell lines have been reported for the combination of simple coumarins with sorafenib (Figure 1) [25,26].

The distribution of coumarins in nature is ubiquitous. Coumarins may be found in plant roots, seeds, leaves, flowers, peels and fruits, as secondary metabolites. Their extraction and isolation from various plant species has been extensively studied due to their relevant biological activity in many therapeutic areas [27]. Some coumarin structures have also been identified in marine sources, especially in algae, marine fungi and ascidians. As shown before, new bioactive molecules from oceans could facilitate the discovery of chemical analogues with potential applications in drug discovery. For this reason, the purpose of this review is to exclusively cover coumarin derivatives from marine sources with a pharmacological interest, as well as the novel or fruitful synthetic routes described to obtain these privileged scaffolds.

## 2. Coumarin Derivatives from Marine World: Synthesis and Activities

The biological properties of coumarin derivatives have been studied in a variety of fields. The great interest that this family of compounds has attracted along the years is reflected in the number of research and review papers dedicated to this topic on PubMed (more than 2000 in just 2022). For marine-derived coumarins, their cytotoxic properties are by far the most studied.

Coumarins from oceans have been mostly found in coastal plants, bacteria, mollusks, invertebrates and sponges [10]. However, extraction from natural sources is time-consuming and the amount of isolated compound is usually scarce. In this sense, many efforts have been made to afford fruitful synthetic routes in various operational steps to achieve the desired derivatives. 

In this review, we will simulate showcasing the pharmacological activities and relevant synthetic routes for specific families of derivatives and their analogues. Novel methodologies developed during the last years will also be covered to illustrate new synthetic opportunities.

Although a general classification of coumarin derivatives has been presented in the Section 1, a suitable classification of coumarins from the marine world into different categories would be more appropriate. For simplicity, we will distinguish between simple, 3-substituted (amino- and imino-), tricyclic (benzo- and furo-), pentacyclic and other coumarin analogues. 

### 2.1. Simple Coumarins

Simple substituted coumarins are the structurally less complex class of coumarins. The scaffold is constituted by a bicyclic system and different substitution patterns at the C-3, C-4, C-5, C-6, C-7 and C-8 positions. Due to the already defined biological activity [28,29], two compounds—umbelliferone (**7**, R^7^ = OH) and scopoletin (**8**, R^6^ = OCH_3_ and R^7^ = OH) —are highlighted here (Figure 2). These molecules were isolated in 2012 from the leaves of the mangrove endophytic fungus *Penicillium* sp. ZH16 from the South China sea [30]. 

Umbelliferone (**7**) shows anti-inflammatory [31] and antitumoral activities [32]. In 2015, Yu et al. reported the anticancer activity of umbelliferone (**7**) against HepG2 cancer cells, inducing apoptosis in cells [33], whereas scopoletin (**8**) inhibits PC3 proliferation, a human prostate cancer cell line [34]. Additionally, both compounds exhibit anti-acetylcholinesterase (AChE) [35,36] and antioxidant activities [37].

As a result of the importance of these scaffolds in organic and medical chemistry, many synthetic routes to obtain simple substituted coumarins have been explored over the years [38,39]. The most classical strategies involve Knoevenagel [40], Pechmann [41] and Perkin [42] condensations, intramolecular [43,44] and intermolecular Wittig reactions [45], ring-closing metathesis [46], as well as different reactions between the corresponding salicylaldehydes with ketene [47] or arylacetonitriles (**18**) [48] (Figure 1).

Over the last few years, transition metal catalysis, involving palladium [49,50], rhodium [51], iron [52] or cobalt [53], has also been used to synthesize different coumarin derivatives. In addition, modern methodologies such as microwave irradiation [54,55], flow chemistry [56], photochemistry [57], ionic liquids [58,59] and organocatalytic reactions [60], proved to be very effective.

For instance, Y. Li et al. employed the photocatalytic isomerization of *ortho*-*E*-hydroxycinnamates (**19**) to generate *Z* isomers, which underwent lactonization to provide coumarin compounds in high yields (Figure 2) [61]. A similar strategy was developed by Shu et al. but using Ir_2_(ppy)_4_Cl_2_ as the catalyst (Figure 2) [62].

Au(I)-catalysts have also been screened to synthesize coumarins by the intramolecular arylation (IMHA) of phenol-derived propiolates (**20**) [63]. IMHA reactions were carried out using Echavarren’s catalyst (**21**), (acetonitrile)[(2-biphenyl)di-*tert*-butylphosphine]gold(I) hexafluoroantimonate, to give numerous derivatized compounds in high yields (Figure 3).

Furthermore, other metal-free methodologies have been reported. In 2016, Lee et al. reported a TfOH-mediated condensation of phenols (**22**) with propiolic acids (**23**), followed by intramolecular arylation [64], which was applied to obtain natural umbelliferone (**7**) in an 81% yield (Figure 4).

### 2.2. 3-Substituted (Imino- and Amido-) Coumarins

#### 2.2.1. 3-Iminocoumarins

All 3-iminocoumarins (**24**–**39**) reported in the literature have been isolated from mangrove fungi present in the South China sea, along with tens of other metabolites [65]. To the best of our knowledge, these compounds do not show relevant biological activity, and no synthetic approaches have been published to date (Figure 3).

#### 2.2.2. 3-Amidocoumarins

Trichodermamide A (**40**), B (**41**) and aspergillazine A (**42**) were the first 3-amidocoumarins isolated at the beginning of the 21st century from different marine-derived fungi, *Trichoderma virens* and *Spicaria elegans* [66,67,68]. Spectroscopic analysis and chemical methods (a modified Mosher’s method) allowed for the determination of the structure and stereochemistry of **40** [66] and **42** [69], while the structure of **41** was established by X-ray diffraction analysis (Figure 4) [66].

Compounds **40** and **42** proved to display a weak cytotoxic activity against an HL-60 cell line (IC_50_ = 89 and 84 Μm, respectively) [67]. By contrast, as a result of the presence of the chlorohydrin group in **41**, it displays in vitro cytotoxicity against HCT-116 colorectal cancer cells (IC_50_ = 0.32 μg/mL) [66] and HeLa cells (IC_50_ = 3.1 μM), by breaking double-stranded DNA [70]. Moreover, weak antimicrobial activities have been reported [68].

Recently, four new 3-amido compounds have been isolated (Figure 5). On the one hand, long-term static fermentation of the strain of *Trichoderma* sp. TPU199 (cf. *Trichoderma brevicompactum*) induced the production of dithioaspergillazine A (**43**), which possesses a disulphide bridge by comparison with spectroscopic data. In contrast to aspergillazine A (**42**), the compound inhibits the proliferation of the colon cancer HCT-15 cell line (IC_50_ = 13 μM) and Jurkat leukemia cells (IC_50_ = 1.3 μM) [71]. On the other hand, trichodermamide C (**44**) and hatsumamide A (**45**) and B (**46**) were isolated from the deep sea-derived fungus *Penicillium steckii* FKJ-0213 by physicochemical (PC) screening [72]. The structure of **44**, which also contains a 1,2-oxazine system, was previously established by NMR, UV, IR, MS and X-ray diffraction data. It shows moderate cytotoxic effects towards human colorectal carcinoma HCT116 (IC_50_ = 0.68 μg/mL) and human lung carcinoma A549 (IC_50_ = 4.28 μg/mL) [73]. The structure and stereochemistry of **45** and **46** were elucidated by mass spectrometry, 1D and 2D NMR data (COSY, HMQC, HMBC and ROESY) and by comparing data with other already known compounds. No biological activity of **46** has been reported. However, **45** presents antimalarial activity against the K1 and FCR3 strains of *Plasmodium falciparum*, with IC_50_ values of 27.2 an 27.9 μM, respectively, and cytotoxicity against five human tumor cell lines, HeLa S3 (IC_50_ = 15.0 μM), HT29 (IC_50_ = 6.8 μM), A549 (IC_50_ = 13.7 μM), H1299 (IC_50_ = 18.7 μM) and Panc1 (IC_50_ = 12.9 μM) [72].

Trichodermamides **40**, **41** and **44** could be disconnected into two fragments: an oxazine ring moiety **47** and an aminocoumarin **48** (Figure 5). Different synthetic strategies have been developed in order to afford the 4*H*-5,6-dihydro-1,2-oxazine fragment **47**.

In 2008, Joullié and Wan described the total synthesis of **40** and **41** [74]. Thus, the advanced intermediate **52**, obtained in 18 linear steps from (–)-quinic acid **49**, was treated with hydroxylamine to obtain an oxime, which in situ underwent an intramolecular epoxide ring opening upon addition of NaOH. Oxazine **53**, obtained as a single diastereomer, was then converted (over four reaction steps) into enone **54**. A Luche reduction followed by selective primary alcohol oxidation provided acid **55** in a good yield (Figure 6).

The coupling reaction between carboxylic acid **55** and aminocoumarin **B** (obtained in three steps from 3,4-trimethoxy-benzaldehyde) was performed using EDCI in 30% pyridine/dichloromethane. Compound **40** was obtained after coupling and two deprotection steps in a 53% yield, while **41** required an additional treatment with mesyl chloride in order to obtain the corresponding allylic chloride (32% over four reaction steps). The total enantioselective syntheses were achieved in 31/32 reaction steps, with an average of an 85% yield for each reaction step (Figure 7) [74].

More recently, a new concise total synthesis of trichodermamides A, B and C has been described [75]. A 1,2-addition of an α*C*-lithiated *O-*silyl ethyl pyruvate oxime **57** to benzoquine **58**, followed by an oxa-Michael ring closure was applied to accomplish the formation of the *cis*-fused 1,2-oxazadecaline core **59** in a 92% yield. A modified Luche reduction and treatment with Pd(PPh_3_)_4_ in the presence of *N*,*O*-bis-(trimethylsilyl)acetamide (BTSA) was carried out to give dienol **60** in a high yield, a common intermediate in the synthesis of the three natural products (Figure 8).

Once the oxazadecaline **60** was obtained, similar chemical steps, but arranged in different order, provided the three natural compounds. In these syntheses, the two key steps are the amide coupling, mediated by HATU in the presence of *sym*-collidine, and a final selenoxide [2,3]-sigmatropic rearrangement with H_2_O_2_ in pyridine [76], which was previously used by Zakarian and Lu [77]. Trichodermamides **40**, **41** and **44** have been obtained after 9, 12 or 13 steps, respectively, in moderate yields (Figure 9).

To the best of our knowledge, no synthetic approach for **42**, **43**, **45** and **46** has been reported yet.

### 2.3. Tricyclic Coumarins

#### 2.3.1. Furocoumarins

First, furocoumarins (also called psoralenes) are described as coumarin derivatives with a fused furan ring with important biological activities, such as photoreactivity with DNA [78]. Four structures of furo[*g*]coumarins (**66**–**69**) were found in the endophytic fungus *Penicillium* sp. ZH16 from the South China sea (Figure 6) [30]. 

From this series, the derivative **68** was tested against KB and KBV200 cells demonstrating relevant cytotoxicity (IC_50_ of 5 and 10 μg/mL, respectively). In addition, **69** shows interesting photochemotherapeutic effects under near UV and blue light photosensitization (LD_50_ of 2 nM and 12 nM, respectively). These cytotoxic studies suggest the possibility of furocoumarins being involved in the high incidence of cancer in Nigeria [79].

The synthesis of furocoumarins has been known for a long time. However, novel methodologies have been developed during the last decades for the effective preparation of these compounds, many of which involve metal-catalyzed transformations that provide new structural diversity [80,81]. 

One such example has been the preparation of bergaptene (**67**). Although it was first isolated in 1834, it was not until 1936 when the first synthetic approach was described. Then, other methodologies were developed during the next decades [82]. Recently, Zhimin et al. reported a new synthesis of **67** in a higher isolated yield (55%) compared to the reported methodologies. Following known methodologies, phloroglucinol was used as the starting material to construct benzofuran-3-one **72** (by monomethylation and Pechmann reactions). Various conventional steps (acetylation, deacetylation) provided the intermediate **75** from which a fused lactone ring was constructed by acetylation and Pechmann condensation. A final dehydrogenation step with DDQ provided the final product **67** (Figure 10).

#### 2.3.2. Benzo[c]coumarins

Benzocoumarins are π-extended structures in which the coumarin core is fused with a benzene ring at different positions. Four benzo[*c*]coumarins (**77**–**80**, 3,4-benzocoumarins or alternariol derivatives) have been found from ocean sources, produced by the mangrove endophytic fungus No. 2240, from the South China sea coast (Figure 7) [83]. The structures were determined by spectroscopic analysis using NMR, IR and UV experiments and by comparison with the literature data. 

Alternariol (**77**, R^1^ = R^2^ = OH) and its derivatives (**78**–**80**) have demonstrated mutagenic properties against the human epidermoic carcinoma KB and KBv200 cell lines. In particular, **77** and **80** show stronger IC_50_ values (3.17, 3.12 and 4.82, 4.94 μg/mL, respectively) in both cell lines, in comparison with the weaker activities found for the other compounds (IC_50_ > 50 μg/mL).

The construction of benzocoumarins depends on the location of hydroxyl and formyl groups on the starting material, which is normally hydroxynaphthaldehyde. Many complementary strategies have been reported for the general synthesis of benzo[*c*]coumarin derivatives, which are based on carbon–carbon and carbon–oxygen bond formation strategies or cyclization reactions [84].

A total synthesis of **77** was described independently by the Podlech and Kim groups [85,86]. The key step in both protocols was a Suzuki–Miyaura cross-coupling of boronic acid **81** with a brominated aldehyde **82** to obtain an advanced precursor **83**. A final cyclization step was needed to obtain the final product **77** (Figure 11).

More recently, the most common techniques in the synthesis of general benzo[*c*]coumarins have been the oxidative cyclization of biphenyl-2-carboxylic acid compounds [87] and Hurtley condensation [88]. However, other strategies that generate chemical diversity have been applied to obtain highly functionalized benzco[*c*]coumarins. For instance, Bodwell et al. prepared a set of benzo[*c*]coumarins by an inverse electron demand Diels–Alder reaction [89]. Later on, a multicomponent version (**9**, **84** and **85**) comprising 6 reaction steps and increasing chemical diversity was disclosed by the same groups (Figure 12) [90]. 

Later, Lee’s group reported the reaction of hydroxychalcones (**86**) and β-ketoesters (**87**), in the presence of a base, in sequential Michael addition/intramolecular aldol condensation/oxidative aromatization/lactonization processes (Figure 13) [91].

### 2.3.3. Other Tricyclic Coumarins

In addition to the previously mentioned groups, other tricyclic coumarins have been found in marine organisms. The limited number of their structural features does not allow for their classification in a particular group. Two pyrano[*g*]coumarins were isolated from *Streptomyces violans* bacteria and *Ascomycete Leptosphaeria oraemaris* fungi (compounds **87** and **88**, respectively). In addition, iotrochotazine A (**89**) was found in the marine sponge *Iotrochota* sp. in Australia, and it is used as chemical probe to study Parkinson’s disease [92]. A series of dihydrocoumarins (**90**–**95**) was also found in *Rhizophora stylosa* mangrove trees in Okinawa, Japan. These compounds present DPPH free radical scavenging properties (EC_50_ 4.6–10.3 μM) and serve as traditional medicine for the local people due to their antioxidant activities (Figure 8) [93,94].

To the best of our knowledge, there are not many protocols reported for the preparation of each of these scaffolds. For instance, cinchonain derivatives **92** and **93** were successfully synthetized by the Kadota group, in a one-pot regioselective procedure involving a dienone–phenol rearrangement followed by a Michael-type reaction (Figure 14) [95].

Considering **89**, a total synthesis was developed in 2014 through a one-pot enamine formation/intramolecular conjugate addition/oxidation sequence. The confirmation of the natural product allowed for subsequent biological investigations (Figure 15) [92].

## 2.4. Pentacyclic Coumarins

### 2.4.1. Aflatoxins

These heterocyclic compounds are difurocoumarin derivatives consisting of a highly reactive bifuran ring and a five-membered lactone fused to the coumarin nucleus. Different aflatoxins (**101**–**106**) were found endogenously in *Aspergillus flavus* 092008 and in different algae in Putian Pinghai, China [96,97,98] (Figure 9).

Aflatoxins are toxic and described as among the most carcinogenic substances. They can be present as contaminants in food. Once in the body, they pass through the liver to generate a reactive epoxide intermediate, or they are hydroxylated to generate the less harmful aflatoxin M1. On the other hand, **101** has been analyzed against A549, K562 and L-02 cell lines, and a weak activity has been described (IC_50_ values 8.1, 2.0 and 4.2, respectively).

Due to the important implications of aflatoxins in human health and the problems associated with their extraction from natural sources, much progress has been made towards the total synthesis of both its racemic and asymmetric versions [99]. 

Outstanding contributions in the total synthesis were carried out by the Büchi group, such as the first total synthesis of **101** in a racemic version [100]. Considering this derivative as an example, an asymmetric total synthesis was developed for the first time by Trost et al. [101]. The synthesis of the key precursor **107** was carried out via a Pechmann reaction starting from 5-methoxy-*m*-catechol and β-keto ester. Then, the chiral center of the B ring was achieved in a high yield and *ee* through an allyl–palladium intermediate. Once synthetized, an intramolecular Heck reaction under standard conditions gave the new BC ring. A similar strategy allowed the construction of the final ring, which through acylation and pyrolysis generated the desired aflatoxin **101** (total yield in 9 steps of 1.6%) (Figure 16).

### 2.4.2. Lamellarins

Most commonly, most of the coumarin derivatives extracted from marine sources correspond to lamellarins. This group of alkaloids, with an unprecedented chemical structure in the natural world, has been found in a plethora of marine invertebrates (sponges, tunicates and mollusks). Lamellarins A–D were first isolated in 1985 from the mollusk Lamellaria sp. in Palau, and over other 70 lamellarins have been isolated since then [102]. As shown in Figure 10, their structure contains a 3,4-diarylpyrrole moiety, similarly to the previously mentioned ningalin derivatives [103]. 

With respect to their biological activity, many lamellarins have been tested in a number of cancer cell lines. Thus, important anticancer activities have been described with IC_50_ values from the nanomolar to the micromolar ranges. In this series, lamellarins D (**113a**, R^1^ = R^4^ = R^6^ = R^7^ = H, R^2^ = R^3^ = R^5^ = Me), M (**113b**, R^1^ = R^4^ = H, R^2^ = R^3^ = R^5^ = R^6^ = Me, R^7^ = OH) and K (**114a**, R^1^ = R^4^ = R^8^ = H, R^2^ = R^3^ = R^5^ = R^6^ = Me, R^7^ = OH) may be highlighted because of their cytotoxicity (38–110 nM). Other analogues such as lamellarin N (**113c**, R^1^ = R^3^ = R^6^ = R^7^ = H, R^2^ = R^4^ = R^5^ = Me), X (**113d**, R^1^ = R^3^ = H, R^2^ = R^4^ = R^5^ = R^6^ = Me, R^7^ = OH) and J (**114b**, R^1^ = R^6^ = R^7^ = R^8^ = H, R^2^ = R^3^ = R^4^ = R^5^ = Me) proved to be even more potent and could be potential candidates for an anticancer drug discovery program [1]. 

In addition, lamellarin D (**113a**) is able to inhibit the enzyme topoisomerase I, and it has been tested against human prostate and leukemia cell lines [104]. Finally, a series of lamellarins has been tested in vitro against colorectal cancer cells (COLO-205) with IC_50_ values up to 0.0056 μM [105]. Other natural analogues have allowed for SAR studies, by demonstrating the interaction with P-glycoprotein (P-gp) and opening new avenues in the development of non-cytotoxic P-gp inhibitors for human colon cancers [106]. 

For decades, many efforts have been made to develop efficient synthetic routes for these structures [107]. Within the strategies most recently described for the construction of the skeleton, we can find 1,3-dipolar cycloaddition [108], aza-Nazarov reactions [109] or Grob-type coupling [110]. 

Okano et al. also described the total synthesis of some lamellarins using a one-pot lithiation/Negishi coupling methodology [111,112]. In addition, novel technologies such as photocatalysis and electrosynthesis also proved to be effective in the construction of the core [113]. For instance, a photocatalytic tungsten-catalyzed [3 + 2] cycloaddition reaction provided a family of pyrrolo [2,1-*a*]isoquinolines (Figure 17) [114].

In general, most approaches are based on the functionalization of the commercially available pyrrole enroute to the natural product or the construction of a functionalized pyrrole to continue the synthesis. In any case, there is still no reported chemical synthesis for some of these derivatives and some retrosynthetic efforts have been made to increase their diversity.

Khan’s group demonstrated a scalable total synthesis of lamellarins S, Z, G, L, N and D in 5 or 6 steps (20–27% of overall yield). A retrosynthetic analysis was based on a double Pd-mediated cross dehydrogenative coupling (CDC) which would construct the desired core after oxidation. Then, a coupling of the carboxylic acid **120** with phenols would account for the introduction of the new ring. Finally, a [3 + 2] cycloaddition strategy of aziridine-2-carboxylates (**121**) and β-bromo-β-nitrostyrenes (**122**) would afford the desired pyrrole ring (Figure 18) [115].

## 2.5. Ningalins

Ningalin derivatives contain one or two coumarin scaffolds fused to a pyrrol ring. Some of these structures have been found from marine sources, containing penta- and tricyclic scaffolds. Particularly, ningalins A (**123**), B (**124**), F (**125**) and E (**126**) have been isolated from the ascidian *Didemnun* and/or sponge *Iotrochota baculifera* in the Ningallo reef region in Australia, giving the name to this family (Figure 11) [116,117]. One of the most remarkable activities of ningalin analogues is their ability to suppress HIV replication. In this regard, the structures in Figure 11 have been able to act as potent inhibitors against the HIV-1 IIIB virus in both MT4 and MAGI cell lines [117,118]. In addition, they also show potent inhibition against kinases related with neurodegenerative diseases, such as cyclic-dependent kinase 5 (CDK5), glycogen synthase kinase 3b (GSK3b) and casein kinase I (CD1d) [116].

To illustrate the synthetic approaches of this family, we chose **124** as a representative example. In this sense, many of the most recent strategies developed have been based on describing a new synthetic methodology and applying it to the preparation of the ningalin. For instance, Okano et al. developed a convergent total synthesis starting from a dibromopyrrole derivative [111]. A double Suzuki–Miyaura coupling generated the corresponding diarylated pyrrole **129**. Then, removal of the SEM group via an intramolecular cyclization step gave rise to the tricyclic analogue **130**. The desired product **124** was obtained under Mitsunobu conditions and hydrogenolysis, in a 97% yield (Figure 19).

Moreover, a one-pot multistep methodology for the construction of this core was reported in 2019 by Yang and colleagues. This methodology, comprising coupling, hydrolysis, reduction and cyclization steps, generated the derivative **134**. Subsequent *N*-alkylation in the presence of a base and final demethylation gave rise to **124** with an overall yield of 42% (Figure 20) [119].

## 2.6. Other Coumarin Derivatives: Bicoumarins and Tetracyclics

Many of the coumarin derivatives already shown are constituted by 3- or 5- fused cycles and have been classified considering this criterion. Nevertheless, other scaffolds have been found in different types of *Aspergillus* in Australia, which have been identified as dimers of coumarin, demethylkotanin (**136**) and bicoumanigrin (**137**) or with a tetracyclic core, aspergiolide A (**138**) and B (**139**) [120,121,122] (Figure 12). The antiproliferative activity and pharmacokinetic properties of **138** and **139** have been measured against several cancer cell lines. The best results have been displayed in A-549 and HL-60, with IC_50_ = 0.13 and 0.28 μM for **138**, and IC_50_ = 0.24 and 0.51 μM for **139**, respectively. Moreover, an in silico analysis of aspergiolide B (**139**) indicated a low binding free energy in the active site, which could be a potential EGFR-TK inhibitor [123].

To the best of our knowledge, the total synthesis of **138** and **139** has not yet been described. However, Li, Liu and colleagues developed a strategy for the preparation of the main core in 2019 [124]. Moreover, a library of simplified analogues was synthetized and evaluated against various cell lines. The core structure (**143**) was constructed in a two-step procedure based on the Knoevenagel condensation and transesterification sequence, followed by an intramolecular Friedel–Crafts acylation. This protocol implies an advance in the underdeveloped synthesis of aspergiolides (Figure 21).

## 3. Conclusions

The ocean is still an underexplored world in the search for active molecules around Earth’s different environments. Here, coumarin derivatives have been identified in a variety of living organisms. They present a diverse set of structural features which confer interesting biological properties that have been tested in a plethora of cell lines and other enzymatic assays. Particularly, some of these derivatives and analogues have been highlighted because of their antitumor, antioxidant, antimicrobial and/or enzymatic inhibitory properties. For instance, psoralene and alternariol derivatives show promising cytostatic and photochemotherapeutic properties against KB and KBV200. Ningalins act as potent inhibitors against the HIV-1 virus and kinases related with neurodegenerative diseases (CDK5, GSK3b, CD1d). Lamellarins have been tested in a number of cell lines, demonstrating cytotoxic and antitumor effects, and being the most interesting due to their pharmacological properties. Although the extraction of coumarin derivatives from oceans is tedious, and low quantities are normally obtained, many synthetic efforts have been made to achieve efficient synthetic methodologies. Classical strategies involving Knoevenagel, Pechmann and Perkin condensations, as well as Wittig or ring-closing metathesis, have been used to construct the coumarin core. The total syntheses of trichodermamides, aflatoxins and lamellarins have been extensively carried out during the last decades. Moreover, transition metal catalysis, photocatalysis or even multicomponent approaches have been used as convenient strategies to expand the chemical space of coumarin derivatives. We envision that chemical modifications of marine-derived coumarins may improve their pharmacological properties and provide a step forward in the development of new therapies.

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
