# Peer review of "Recent Advances in Biologically Active Coumarins from Marine Sources: Synthesis and Evaluation"

_marinedrugs, 2022, doi:10.3390/md21010037_

Round 1

Reviewer 1 Report

The review discusses the most interesting coumarins of marine origin from the point of view of medicinal chemistry, and describes new and useful synthetic approaches to coumarin derivatives.

The review presents interesting data in the important area of research and demonstrates excellent knowledge of the material. I recommend this review for publication.

However, the presentation in the Conclusions section is too general. In my opinion, some analysis of the data on the synthesis and properties of coumarins presented in the review deserves to be included in this section in a concise form.

Author Response

Thank you to reviewer 1 for the comments that have helped to improve the quality of the manuscript.

Specific data has been added to conclusions section, showing the most promising coumarin derivatives as well as relevant synthetic methodologies.

Reviewer 2 Report

The authors summarized the marine sourced coumarins, including their diverse structure feathers and synthetic approaches. The review is well-organized and would be a good reference resource for those who are interested in coumarin chemistry. 

Minor suggestions:

1, Delete the revision comment on lines 47 and 145

2, The scale of the chemical structure isn’t consistent. For example, on page 11, the structures in figure 7 are larger than scheme 11. Same issues in pages 13, 17, and 18. The authors should keep using the same scaling for structures on the same page.

Besides the minor things above,  I recommend its publication in Marine Drugs.

Author Response

Thank you to reviewer 2 for the comments that have helped to improve the quality of the manuscript.

Revision comments have been deleted and chemical structures have been scaled